# Understanding disruptions in cancer care to reduce increased cancer burden

**Kia L Davis**[1]*, **Nicole Ackermann**[1], **Lisa M Klesges**[1], **Nora Leahy**[1], **Callie Walsh-Bailey**[2], **Sarah Humble**[1], **Bettina Drake**[1], **Vetta L Sanders Thompson**[2]

[1]Department of Surgery, Public Health Sciences, School of Medicine, Washington University in St. Louis, St Louis, United States; [2]Brown School, Washington University in St. Louis, St Louis, United States

## Abstract

**Background:** This study seeks to understand how and for whom COVID-19 disrupted cancer care to understand the potential for cancer health disparities across the cancer prevention and control continuum.

**Methods:** In this cross-sectional study, participants age 30+residing in an 82-county region in Missouri and Illinois completed an online survey from June-August 2020. Descriptive statistics were calculated for all variables separately and by care disruption status. Logistic regression modeling was conducted to determine the correlates of care disruption.

**Results:** Participants (N=680) reported 21% to 57% of cancer screening or treatment appointments were canceled/postponed from March 2020 through the end of 2020. Approximately 34% of residents stated they would need to know if their doctor's office is taking the appropriate COVID-related safety precautions to return to care. Higher education (OR = 1.26, 95% CI:1.11–1.43), identifying as female (OR = 1.60, 95% CI:1.12–2.30), experiencing more discrimination in healthcare settings (OR = 1.40, 95% CI:1.13–1.72), and having scheduled a telehealth appointment (OR = 1.51, 95% CI:1.07–2.15) were associated with higher odds of care disruption. Factors associated with care disruption were not consistent across races. Higher odds of care disruption for White residents were associated with higher education, female identity, older age, and having scheduled a telehealth appointment, while higher odds of care disruption for Black residents were associated only with higher education.

**Conclusions:** This study provides an understanding of the factors associated with cancer care disruption and what patients need to return to care. Results may inform outreach and engagement strategies to reduce delayed cancer screenings and encourage returning to cancer care.

**Funding:** This study was supported by the National Cancer Institute's Administrative Supplements for P30 Cancer Center Support Grants (P30CA091842-18S2 and P30CA091842-19S4). Kia L. Davis, Lisa Klesges, Sarah Humble, and Bettina Drake were supported by the National Cancer Institute's P50CA244431 and Kia L. Davis was also supported by the Breast Cancer Research Foundation. Callie Walsh-Bailey was supported by NIMHD T37 MD014218. The content does not necessarily represent the official view of these funding agencies and is solely the responsibility of the authors.

*For correspondence: DavisKL@wustl.edu

## Editor's evaluation

The study presents patterns of cancer care disruption in southern Illinois and eastern Missouri in the summer of 2020. Survey results show factors that impact cancer care during the COVID-19 pandemic, including group differences by race. The important findings provide solid evidence about variation in cancer care disruptions and opportunities to improve return to care.

## Introduction

The COVID-19 pandemic abruptly upended cancer care in many countries including the US. The need to reduce community spread and reserve hospital capacity for the most severe COVID-19 cases led to rescheduling or postponement of cancer care appointments (*Nelson, 2020*; *Patt et al., 2020*; *Ueda et al., 2020*; *Zheng et al., 2021*; *Wenger et al., 2022*). These control measures significantly decreased cancer-related patient encounters in the early phase of the pandemic, particularly for cancer screenings (*Patt et al., 2020*). Comparing March to July 2020 with the same period in 2019, there was a substantial decrease in cancer screenings, biopsies, surgeries, office visits, and therapy; the decreases varied by service location and cancer type (*Patt et al., 2020*). For example, breast cancer screenings decreased by 89.2% and colorectal by 84.5% (*Warner et al., 2020*). Patients reported delays in receiving cancer care, including follow-up clinic appointments and cancer therapies, such as radiation, infusion therapies, and surgeries (*London et al., 2022*; *Riera et al., 2021*).

Cancer care delays due to the COVID-19 pandemic are anticipated to lead to increased cancer morbidity and mortality (*Blay et al., 2021*; *Malagón et al., 2022*). One study found an association between surgical and screening delays and increased cancer mortality among patients diagnosed with colorectal, lung, and prostate cancer during the pandemic (*Zheng et al., 2021*). Delayed mammography and computed tomography for lung cancer were associated with advanced stage of cancer at diagnosis (*Zheng et al., 2021*). Another study determined delayed surgery for lung cancer was associated with worse survival (*Mayne et al., 2021*). For breast screenings, some evidence suggests that patients were reluctant to return for mammograms after care disruptions (*Miller et al., 2021*). Thus, cancer care disruptions during COVID-19 could have detrimental future impacts on cancer outcomes and may require changes to public health and clinical strategies across the cancer prevention and control continuum.

It is unclear if patients felt comfortable returning to care in the context of rapidly changing information and guidelines related to COVID-19 and even now that guidelines are more consistent and vaccines are available. There is concern about whether patients will prioritize immediate unmet social needs that might be a result of or exacerbated by COVID-19, such as food insecurity, employment loss, and housing challenges, over disease prevention. Furthermore, people of color, including African Americans, Latinx, and Native communities, as well as those employed in low-wage occupations, are likely to have greater concerns over COVID-19 safety, in addition to the immediate concerns noted above (*Cancino et al., 2020*). Rural communities that already experience limited access to cancer care, have less capacity to manage COVID-19 (*Segel et al., 2021*). Finally, hospitals rapidly increased the use of telehealth to continue cancer care during COVID-19, but older people and those who lived in low-income and rural areas, or were less likely to have commercial insurance were less likely to participate (*Darcourt et al., 2021*; *Jaffe et al., 2020*). This combination of factors may exacerbate existing disparities (*Cancino et al., 2020*).

This survey study was conducted by National Cancer Institute (NCI)—designated Siteman Cancer Center to elucidate: (1) to what extent cancer care appointments (including preventive screenings and treatment) in the bi-state Midwestern catchment area were canceled/postponed, (2) patients' needs for returning to care, and (3) correlates of care disruption across the catchment area. This study aligns with the NCI's goal to support population health assessments of their cancer center's catchment areas. In our catchment area, the cancer burden is significantly greater than the US averages for multiple cancers. Moreover, racial and geographical disparities persist such that African American patients have higher incidence and mortality for lung, colorectal, late-stage breast cancer diagnoses, and prostate cancers compared to White patients. Rural counties also have higher mortality (but not incidence) for melanoma, breast, and prostate cancer compared to urban areas (*National Cancer Institute, 2022*).

Thus, we explore how socio-contextual factors impact cancer health disparities across the continuum of cancer control and prevention during COVID-19 in this bi-state Midwestern catchment area. This analysis is guided by the theory that social identities like race, ethnicity, social class, and gender shape many contextual factors related to cancer, COVID, and other outcomes and are ultimately the fundamental drivers of disease. We stratify our results by race because of the differential impact of COVID-19 on communities of color and the over-representation of socioeconomic factors such as low-income, low-wage work often experienced by communities of color (*Acosta et al., 2021*; *Athavale et al., 2021*; *Millett et al., 2020*).

## Methods

### Data source

Data were collected from June through August 2020 as part of Siteman Cancer Center's Community Outreach and Engagement efforts. The survey focused on understanding cancer prevention and control behaviors throughout the Siteman catchment area. The Siteman catchment area includes 82 counties throughout Missouri (40) and Illinois (42) and is diverse concerning race (21% people of color), geography (15% rural), and healthcare access (29% live in medically underrepresented areas as designated by the Health Resources and Services Administration, HRSA) (*United States Census Bureau, 2022*).

### Data collection

The Washington University in St. Louis, MO Institutional Review Board approved and exempted this study (ID#202006089). We recruited participants through Qualtrics Online Panels, which emailed potential participants a survey link (*Qualtrics, 2020*). We screened potential participants for the following eligibility criteria: age 30 or older and residing in eastern or southeastern Missouri or central or southern Illinois. Recruitment oversampled for males (35%), people of color (35%) (defined as all races and ethnicities except for non-Hispanic White), and non-metro area residents (20%) (defined as a score of 4 or greater for census-designated rural-urban continuum [RUCC] codes) (*United States Department of Agriculture, 2019*) to allow for analyses by these groups. The median survey completion time was 20.3 min. All participants received an agreed-upon incentive from Qualtrics.

### Measures

#### Outcome variable

*Supplementary file 1* provides detailed information about the measures used in this study (*Centers for Disease Control and Prevention, 2018*; *Qi et al., 2019*; *Wadhera et al., 2019*; *Wadhera et al., 2020*). Our outcome of interest, care disruption, was defined as any cancelation or postponement of a general medical or cancer screening appointment. Catchment area residents who reported that they decided not to attend an appointment not already canceled/postponed due to COVID-19 or they or their doctor/clinic postponed any cancer screening (Pap test, stool blood test, colonoscopy, mammogram, or PSA test) appointment were categorized as experiencing care disruption. Questions were drawn from validated measures assessing the impact of major life disruptions such as natural disasters, and pandemics such as H1N1 (*Saez-Clarke et al., 2023*).

#### Explanatory variables

We included predictor variables that could result in differential access to care due to social stratification: age, race, (*Blake, 2019*) ethnicity, gender identity (*Killermann, 2020*), sex assigned at birth, sexual orientation, education (*Blake, 2019*), income (*Blake, 2019*), residence in non-metro area, pre-COVID employment, health insurance status, job loss due to COVID-19 (*Grasso et al., 2020*), and access to a private vehicle. We also assessed self-report healthcare discrimination using a seven-item scale assessing how many times a participant experienced certain kinds of treatment (overall Cronbach's alpha = 0.92; *Peek et al., 2011*). We also controlled for whether they scheduled a telehealth appointment (*Penedo et al., 2020*). All items were adapted from standardized measures, except for sex assigned at birth and access to a private vehicle, which were created by the study team.

We asked if residents participated in a telehealth medical appointment since the COVID-19 pandemic started and whether it was for a general medical appointment or cancer care. While this measure is not directly associated with social stratification, it could be correlated with Internet and other technology access and also predict whether someone was more likely to cancel/postpone a scheduled in-person appointment. Finally, we developed a single item to understand what patients who may have experienced care disruption would need *most* to be able to reschedule the appointment. These options included transportation, time to schedule, and knowing: how they would pay for the appointment, if the doctor's office or clinic was taking appropriate COVID-related safety precautions, if the doctor's office was still open or scheduling appointments, or that they could bring someone with them; we also included an 'other' option with an open-ended response field.

## Analytic procedures

Descriptive statistics were obtained for all variables separately and by care disruption status (any care disruption compared to no disruption). Next, logistic regression modeling was conducted to determine the associations with care disruption across the catchment area. For all analyses, 'prefer not to answer' responses were recoded as missing. We dropped those who reported that canceling/postponing an appointment did not apply to them (n=84) with more males, uninsured people, and those without telehealth appointments reflected in this exclusion. Additionally, there were 17 residents who had missing data and were not included in the logistic regression models. This missing data was due to responding to sex assigned at birth or sexual orientation questions with 'prefer not to answer' or having a missing value on another question included in the model (see footnote on *Table 1* for details). We also used sex at birth and not gender identity in the model due to the near-complete overlap between the two variables and the small sample size for some of the gender-diverse categories (N<6). Additionally, we recoded the job loss variable into the following 3 categories: yes, resident was laid off; no, resident was not laid off; and combined categories of don't know/not sure/prefer not to answer/not applicable. Finally, we conducted a stratified logistic regression analysis to determine if the associations of care disruption among non-Hispanic Black residents differed when compared to non-Hispanic White residents. Metro/non-metro area was excluded from the stratified non-Hispanic Black and non-Hispanic White models due to a small number of non-Hispanic Black residents in non-metro areas. We do not present other race/ethnicity in the race-stratified models due to the small sample size of participants with non-missing variables for the model in this category (N=71).

## Results

### Sociodemographic and care disruption descriptive information

Unadjusted sociodemographic characteristics of this diverse sample of residents from the Siteman Cancer Center catchment area (n=680) are presented in *Table 1*. Residents were 46 years old on average. Compared to our catchment area, this sample had a higher proportion of women (68% vs 51%) and college graduates (38% vs 30%). We also had a higher proportion of people of color (41% vs 21%) and residents who lived in rural areas (28% vs 15%) due to intentional oversampling (*United States Census Bureau, 2022*).

In this sample, approximately 55% of respondents experienced disruption to their scheduled healthcare appointments. Those who experienced care disruption were more likely to be female, have higher levels of educational attainment, have scheduled a telehealth appointment, reported slightly higher levels of discrimination, and been laid off or had to close their own business compared to those who did not experience care disruption.

The characteristics of residents across Missouri and Illinois by race are shown in *Supplementary file 2*.

The number of residents scheduled for a cancer screening appointment or cancer care, and whose appointment was canceled/postponed by the patient or their doctor/clinic is presented in *Figure 1*. There were 480 possible appointments scheduled between March 2020 through the end of 2020 for either a mammogram, pap test, blood stool test, colonoscopy, or PSA test. Appointment cancelations/postponements varied from 21%–57% by screening type. Approximately 53 people in the sample reported having cancer. Among those, 26% reported having to cancel/postpone their cancer-related care. Additionally, in our sample, 25% of residents canceled/postponed a scheduled in-person dental appointment, 31% avoided seeking care in a hospital (e.g. labor and delivery, emergency room, etc.), and 46% of residents canceled/postponed a scheduled in-person general medical appointment (data not presented).

### Patient needs for rescheduling

In addition, we asked participants who experienced any care disruption what they would need most to reschedule their appointments (n=376). The largest proportion of participants said they would need to know if their doctor's office or clinic is taking the appropriate COVID-related safety precautions (33.8%), followed by not needing anything (18.1%). Some participants needed to know if their doctor's office is making appointments for general or routine care (13.3%) or stated they were dealing with other things and not ready to reschedule yet (10.6%). Approximately 8.2%

**Table 1.** Characteristics of residents across Missouri and Southern Illinois by care disruption status (July-August 2020).

| Variable | Category | Total sample (N=680) – N (%) | No care disruption (N=304) – N (%) | Care disruption (N=376) – N (%) |
|---|---|---|---|---|
| | White | 399 (58.7%) | 186 (61.2%) | 213 (56.7%) |
| | Black or African American | 212 (31.2%) | 90 (29.6%) | 122 (32.5%) |
| | Asian/ Native Hawaiian or Other Pacific Islander | 21 (3.1%) | 12 (4.0%) | 9 (2.4%) |
| Race | Other, including multiple groups | 48 (7.1%) | 16 (5.3%) | 32 (8.5%) |
| | Yes | 15 (2.2%) | 5 (1.6%) | 10 (2.7%) |
| Hispanic, Latino/a, or Spanish origin | No | 664 (97.8%) | 299 (98.4%) | 365 (97.3%) |
| | Woman | 464 (68.2%) | 192 (63.2%) | 272 (72.3%) |
| | Man | 206 (30.1%) | 110 (36.2%) | 96 (25.5%) |
| | Transgender / Gender Diverse | 5 (0.7%) | 1 (0.3%) | 4 (1.1%) |
| Gender Identity* | Prefer not to answer | 5 (0.7%) | 1 (0.3%) | 4 (1.1%) |
| | Female | 472 (69.4%) | 193 (63.5%) | 279 (74.2%) |
| | Male | 204 (30.0%) | 110 (36.2%) | 94 (25.0%) |
| Sex assigned at birth* | Prefer not to answer | 4 (0.6%) | 1 (0.3%) | 3 (0.8%) |
| | LGBTQIA+ | 76 (11.2%) | 25 (8.2%) | 51 (13.6%) |
| | Straight or Heterosexual | 590 (86.8%) | 272 (89.5%) | 318 (84.6%) |
| Sexual Orientation | Prefer not to answer | 14 (2.1%) | 7 (2.3%) | 7 (1.9%) |
| | Less than High School or GED | 31 (4.6%) | 17 (5.6%) | 14 (3.7%) |
| | Grade 12 or GED (High school graduate) | 120 (17.7%) | 64 (21.1%) | 56 (14.9%) |
| | Some college, but did not graduate | 159 (23.4%) | 78 (25.7%) | 81 (21.5%) |
| | Associates Degree or Technical School Certification | 111 (16.4%) | 42 (13.9%) | 69 (18.4%) |
| | College 4 years or more (College graduate) | 143 (21.1%) | 63 (20.8%) | 80 (21.3%) |
| Education* | Graduate or professional school | 115 (16.9%) | 39 (12.9%) | 76 (20.2%) |
| | $0 to $9,999 | 57 (8.4%) | 32 (10.6%) | 25 (6.7%) |
| | $10,000 to $14,999 | 53 (7.8%) | 19 (6.3%) | 34 (9.1%) |
| | $15,000 to $19,999 | 36 (5.3%) | 14 (4.6%) | 22 (5.9%) |
| | $20,000 to $34,999 | 105 (15.5%) | 43 (14.2%) | 62 (16.5%) |
| | $35,000 to $49,999 | 110 (16.2%) | 50 (16.5%) | 60 (16.0%) |
| | $50,000 to $74,999 | 121 (17.9%) | 60 (19.8%) | 61 (16.3%) |
| | $75,000 to $99,999 | 91 (13.4%) | 45 (14.9%) | 46 (12.3%) |
| Annual Household Income | $100,000 or more | 105 (15.5%) | 40 (13.2%) | 65 (17.3%) |
| Metro or Non-Metro Area (RUCC codes by ZIP Code) | Metro | 493 (72.5%) | 222 (73.0%) | 271 (72.1%) |
| | Non-Metro | 187 (27.5%) | 82 (27.0%) | 105 (27.9%) |

*Table 1 continued on next page*

*Table 1 continued*

| Variable | Category | Total sample (N=680) – N (%) | No care disruption (N=304) – N (%) | Care disruption (N=376) – N (%) |
|---|---|---|---|---|
| | Employed Full-time | 321 (47.4%) | 148 (49.2%) | 173 (46.0%) |
| | Employed Part-time | 72 (10.6%) | 30 (10.0%) | 42 (11.2%) |
| | Unemployed | 61 (9.0%) | 29 (9.6%) | 32 (8.5%) |
| | Homemaker | 65 (9.6%) | 22 (7.3%) | 43 (11.4%) |
| | Student | 4 (0.6%) | 2 (0.7%) | 2 (0.5%) |
| | Retired | 84 (12.4%) | 42 (14.0%) | 42 (11.2%) |
| | Disabled | 62 (9.2%) | 23 (7.6%) | 39 (10.4%) |
| Employment (pre-COVID) | Self-Employed/Other | 8 (1.2%) | 5 (1.7%) | 3 (0.8%) |
| | Private | 314 (46.2%) | 135 (44.4%) | 179 (47.6%) |
| | Medicare/Medicare + | 126 (18.5%) | 59 (19.4%) | 67 (17.8%) |
| | Medicaid | 120 (17.7%) | 47 (15.5%) | 73 (19.4%) |
| | Other/Unknown | 22 (3.2%) | 12 (4.0%) | 10 (2.7%) |
| Insurance | Currently do not have insurance | 98 (14.4%) | 51 (16.8%) | 47 (12.6%) |
| | Yes | 233 (34.3%) | 85 (28.0%) | 148 (39.4%) |
| Telehealth appointment* | No | 447 (65.7%) | 219 (72.0%) | 228 (60.6%) |
| | Cancer Care | 6 (2.6%) | 0 (0%) | 6 (4.1%) |
| | General Health Care | 218 (94.0%) | 81 (96.4%) | 137 (92.6%) |
| Telehealth appointment type | Both | 8 (3.5%) | 3 (3.6%) | 5 (3.4%) |
| Access to Private Vehicle (own or others) | Yes | 611 (89.9%) | 275 (90.5%) | 336 (89.4%) |
| | No | 69 (10.2%) | 29 (9.5%) | 40 (10.6%) |
| | Yes | 135 (19.9%) | 50 (16.5%) | 85 (22.6%) |
| | No | 423 (62.2%) | 204 (67.1%) | 219 (58.2%) |
| Laid off Job or had to close own business* | Don't Know/Not Sure/Prefer Not to Answer | 15 (2.2%) | 2 (0.7%) | 13 (3.5%) |
| | Not Applicable | 107 (15.7%) | 48 (15.8%) | 59 (15.7%) |
| Variable | | Mean (SD) | | |
| Age | | 46.2 (12.6) | 46.0 (13.3) | 46.5 (12.0) |
| Discrimination [†] | | 1.8 (0.8) | 1.7 (0.8) | 1.9 (0.9) |

Missing values: 1 Hispanic/Latina(a)/Spanish origin; 1 Education; 2 Income; 3 Employment.

*Statistically significant difference (P<0.05; Chi-square or Fischer's test for categorical, t-test or Wilcoxon rank sum for continuous).

[†]Average score of 7 items on a scale of (1) never, (2) once, (3) 2 or 3 times, and (4) 4 times or more; higher scores indicate more discrimination.

stated they needed to have time to reschedule the appointment. All other needs were reported by less than 5% of respondents. Results stayed similar when looking at the highest need for rescheduling by race, except that the third highest need for Black residents was time to schedule the appointment (14.2%), and for White residents, it was dealing with other things and not ready to reschedule yet (14.5%). The top two needs to reschedule appointments were the same across race though the proportion of those who selected each need varied somewhat by race. The top two needs were knowing if their doctor's office or clinic is taking the appropriate COVID-related safety precautions (Non-Hispanic Black: 34.2%; Non-Hispanic White: 33.8%; All Other: 32.7%) and not needing anything for all three racial groups (Non-Hispanic Black: 19.2%; Non-Hispanic White: 18.8%; All Other: 12.2%).

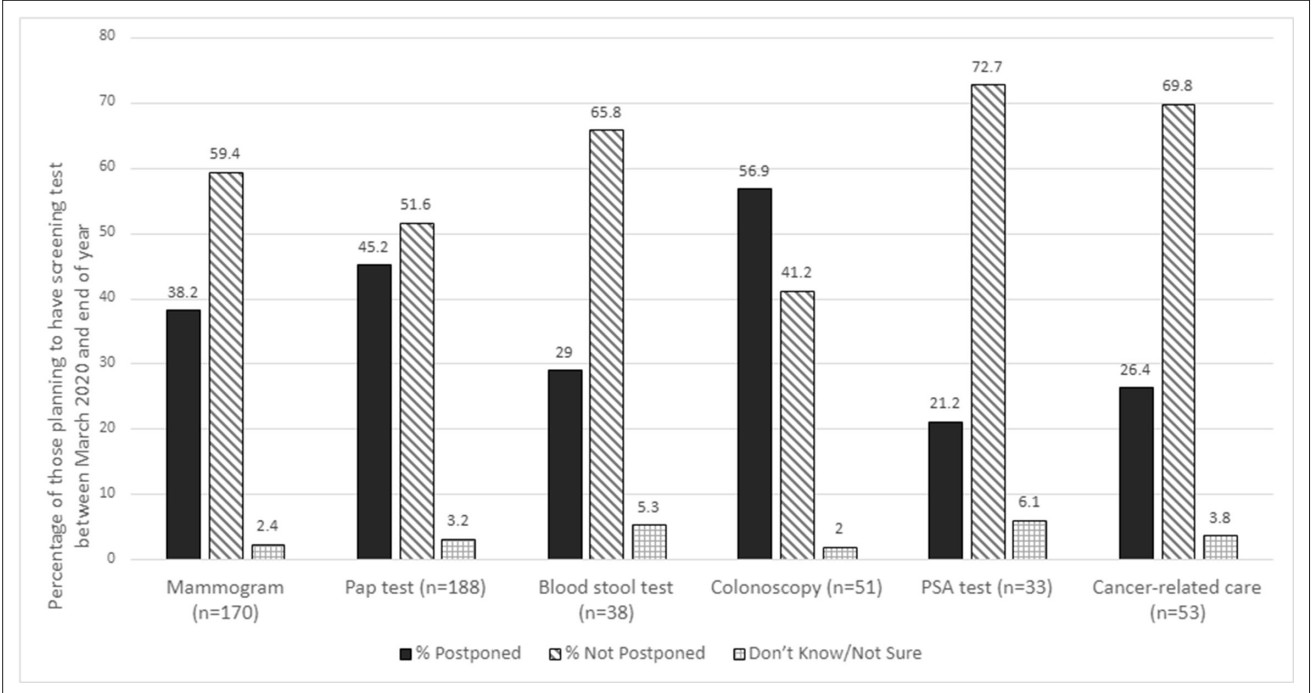

**Figure 1.** Care disruption by cancer screening/appointment type across Missouri and Southern Illinois (July-August 2020). N shown is the number who were planning to have a screening test between March 2020 and the end of 2020; For Cancer-related care, we calculate the percentage out of those who self-reported ever being diagnosed as having cancer (n-53).

## Correlates of care disruption

Logistic regression results for the overall and race-specific models are presented in *Table 2*. In the overall model, higher odds of care disruption were associated with higher educational attainment (OR = 1.26, 95% CI: 1.11–1.43), female (OR = 1.60, 95% CI: 1.12–2.30), reporting experiencing more discrimination in healthcare settings (OR = 1.40, 95% CI: 1.13–1.72), and having scheduled a telehealth appointment (OR = 1.51, 95% CI: 1.07–2.15). The correlates of care disruption were not consistent across race. Among Black residents, only higher levels of educational attainment (OR = 1.45, 95% CI: 1.13–1.85) were associated with greater odds of care disruption. Whereas, among White residents, higher odds of care disruption were associated with higher levels of educational attainment (OR = 1.39, 95% CI: 1.17–1.65), female (OR = 1.90, 95% CI: 1.17–3.08), older age (OR = 1.02, 95% CI: 1.001–1.04), and having scheduled a telehealth appointment (OR = 1.62, 95% CI: 1.01–2.59).

Finally, we included number of comorbidities (classified as 0, 1–2, or 3+comorbidities) in the overall and race-specific models as a posthoc analysis. We found that this variable was statistically significant (p<0.05) in the overall model and the model for Non-Hispanic Black residents. For the Non-Hispanic White model, number of comorbidities is marginally significant (p=0.052). In the overall model, results showed that those with 3+comorbidities were more likely to have care disruptions (vs. 0 comorbidities: OR = 2.01, 95% CI: 1.18–3.45; vs. 1–2 comorbidities: 2.07, 95% CI: 1.31–3.27). Among Black residents, we see similar results – those with 3+comorbidities were more likely to have care disruptions (vs. 0 comorbidities: 2.80, 95% CI: 0.97–8.04; vs. 1–2 comorbidities: 4.82, 95% CI: 1.80–12.93). We take note of the wide confidence intervals and the smaller sample size of Black residents with 3+comorbidities (n=39).

## Discussion

Using primary data collected from residents across the 82-county Siteman catchment area overlapping Missouri and Illinois, we learned that across different type of screenings, 21% to 57% of cancer screening or treatment appointments were canceled/postponed from March 2020 through the end of 2020. Across all races, residents with higher educational attainment had 1.25 higher odds of care

**Table 2.** Odds of any care disruption compared to no care disruption by social factors across Missouri and Southern Illinois (July-August 2020).

| Variable | Overall Sample (N=663) | | Non-Hispanic Black or African American (N=205) | | Non-Hispanic White (N=387) | |
|---|---|---|---|---|---|---|
| | Odds Ratio | 95% CI | Odds Ratio | 95% CI | Odds Ratio | 95% CI |
| Race/Ethnicity | | | | | | |
| Non-Hispanic Black or African American | 1.15 | 0.77, 1.72 | -- | -- | -- | -- |
| Other Race/Ethnicity | 1.40 | 0.79, 2.45 | -- | -- | -- | -- |
| Non-Hispanic White (ref) | -- | -- | -- | -- | -- | -- |
| Sex Assigned at Birth* ‡ | | | | | | |
| Female | **1.60** | **1.12, 2.30** | 1.11 | 0.56, 2.19 | **1.90** | **1.17, 3.08** |
| Male (ref) | -- | -- | -- | -- | -- | -- |
| Sexual Orientation | | | | | | |
| LGBTQIA+ | 1.53 | 0.88, 2.65 | 0.68 | 0.27, 1.72 | 1.65 | 0.73, 3.73 |
| Straight or Heterosexual (ref) | -- | -- | -- | -- | -- | -- |
| Area designation (by ZIP code) | | | | | | |
| Non-Metro | 1.23 | 0.82, 1.84 | -- | -- | -- | -- |
| Metro (ref) | -- | -- | -- | -- | -- | -- |
| Telehealth Appointment* ‡ | | | | | | |
| Yes | **1.51** | **1.07, 2.15** | 1.06 | 0.57, 1.99 | **1.62** | **1.01, 2.59** |
| No (ref) | -- | -- | -- | -- | -- | -- |
| Access to Private Vehicle (own or others) | | | | | | |
| Yes | 0.74 | 0.41, 1.33 | 0.79 | 0.33, 1.90 | 0.74 | 0.27, 1.98 |
| No (ref) | -- | -- | -- | -- | -- | -- |
| Health Insurance | | | | | | |
| Medicare/Medicare + | 0.71 | 0.41, 1.24 | 0.88 | 0.31, 2.47 | 0.75 | 0.37, 1.52 |
| Medicaid | 1.02 | 0.59, 1.75 | 0.76 | 0.32, 1.77 | 1.43 | 0.65, 3.15 |
| Other/Unknown | 0.63 | 0.24, 1.66 | 0.75 | 0.15, 3.92 | 0.38 | 0.09, 1.65 |
| Currently do not have insurance | 0.66 | 0.38, 1.13 | 0.58 | 0.22, 1.52 | 0.87 | 0.42, 1.80 |
| Private (ref) | -- | -- | -- | -- | -- | -- |
| Laid off Job or had to close own business | | | | | | |
| Yes | 1.55 | 0.994, 2.41 | 1.55 | 0.75, 3.20 | 1.52 | 0.80, 2.89 |
| No (ref) | -- | -- | -- | -- | -- | -- |
| Don't Know/Not Sure/Prefer Not to Answer/ Not Applicable | 1.42 | 0.89, 2.25 | 2.12 | 0.87, 5.18 | 1.06 | 0.59, 1.93 |
| Education ‡†* | **1.26** | **1.11, 1.43** | 1.45 | 1.13, 1.85 | 1.39 | 1.17, 1.65 |

*Table 2 continued on next page*

*Table 2 continued*

| Variable | Overall Sample (N=663) | | Non-Hispanic Black or African American (N=205) | | Non-Hispanic White (N=387) | |
|---|---|---|---|---|---|---|
| | Odds Ratio | 95% CI | Odds Ratio | 95% CI | Odds Ratio | 95% CI |
| Income | 0.99 | 0.89, 1.09 | 0.93 | 0.78, 1.13 | 0.99 | 0.87, 1.13 |
| Discrimination* | **1.40** | **1.13, 1.72** | 1.26 | 0.89, 1.78 | 1.29 | 0.96, 1.74 |
| Age[†] | 1.01 | 0.995, 1.03 | 0.99 | 0.96, 1.02 | **1.02** | **1.001, 1.04** |

*Statistically significant (p<0.05) overall variable effect – overall model.

[†]Statistically significant (p<0.05) overall variable effect – Non-Hispanic White model.

[‡]Statistically significant (p<0.05) overall variable effect – Non-Hispanic Black or African American model.

disruption for general or cancer care compared to residents with lower educational attainment; this association remained significant among Black and White residents. Additionally, White residents of older age, assigned female at birth, or having scheduled a telehealth appointment, also had higher odds of care disruption. Interestingly, while the sample size is small, we did see a trend that suggested that Black people with 3+comorbidities were more likely to cancel/postpone care. Finally, knowing their doctor's office or clinic is taking the appropriate COVID-related safety precautions was the greatest reported need for returning to care (33.8%).

Delays in cancer screening can lead to stage shifts where patients are diagnosed at later stages and thus have a higher risk for cancer morbidity and mortality. Understanding which screenings were impacted locally and for whom and identifying patient concerns can inform community outreach and engagement efforts. This allows programs to target groups in their catchment, most likely to have delayed screening and draft messaging that can alleviate patient concerns and in turn facilitate a return to care. Moreover, knowing that those with 3+comorbidities, who likely have additional health needs were more likely to cancel/postpone appointments could also inform recruitment methods in getting people to return to care. It is possible that these cancellations and postponements were due to an increased risk of contracting and dying of COVID; however, it is critical to get this high need population to return to care so widening disparities are avoided.

Mammograms and Pap tests are an area of increased interest for our catchment area given the high number of women scheduled for screening. Approximately 38% of the 170 women who were scheduled for mammograms had canceled/postponed appointments. Similarly, 45% of the 188 women scheduled for Pap tests had canceled/postponed appointments. Delays in colorectal cancer screening impacted a smaller number of people, but colorectal cancer screening is an important area given the high proportion of cancellations/postponements, overall low number of scheduled appointments in general, and high colorectal cancer disparities in the region. Of the 51 people scheduled for a colonoscopy, 57% canceled/postponed appointments, and of the 38 scheduled for a blood stool test, 29% canceled/postponed appointments as well. To help healthcare systems reduce the cancer screening deficit, community outreach and engagement strategies need to address these needs. For example, employing mobile strategies such as the use of mobile mammography and home-based cervical and colorectal cancer screening tests could serve those most impacted.

These data are consistent with prior literature that suggests a reduction in general medical and cancer-related appointments (*Patt et al., 2020*; *Wenger et al., 2022*; *London et al., 2022*; *Czeisler et al., 2020*). This study allows us to understand the magnitude of the impact across eastern/southeastern Missouri and southern Illinois. Future research exploring whether those with higher educational attainment were more likely to cancel/postpone appointments because they were more likely to have better access to scheduling future appointments could further elucidate the extent of educational disparities in healthcare access.

These cross-sectional data cannot infer causality however, many of the correlates of interest (e.g. race, educational attainment) pre-date COVID-19 and the need to consider postponing clinical care. Thus, it is unlikely these results are subject to reverse causation. Also, those excluded due to missing data were more likely to be uninsured. If uninsured persons were also more likely to have canceled/

postponed appointments, this could potentially bias results about care disruptions by insurance status towards the null and underestimate the impact. Finally, unmeasured confounding is possible in cross-sectional studies like ours. Despite these limitations, this is a significant study that can improve our understanding of COVID-19 impacts on cancer prevention and control and offer specific insights into the region. In our data, those with higher education were more likely to cancel/postpone care. This indicates that any trends seen in increasing late-stage diagnosis might occur across socioeconomic categories. Additionally, while Black and White people of higher educational attainment both had increased odds of care disruption, having a scheduled telehealth visit was significantly associated with higher odds of care disruption only for White residents. This suggests that while White people were canceling/postponing in-person care, this care may have been substituted with telehealth appointments. Many of these screenings cannot be done virtually, yet this warrants further investigation to understand if care disruption does not always equate to being disconnected from healthcare for some and the subsequent impact on racial disparities in cancer care.

## Acknowledgements

The authors thank the survey participants for their time, effort, and contribution to the study.

## Additional information

### Competing interests

Lisa M Klesges: has received consulting fees from Dana Farber Cancer Institute. The author is on the Board of Directors for American College of Epidemiology and Neighborhood Preservation, Inc. The author has no other competing interests to declare. Vetta L Sanders Thompson: has received consulting fees from Novaris and Chan-Zuckerburg Initiave via National Academies of Science. The author has received payment or honoraria for lectures, presentations, speakers bureaus, manuscript writing or educational events from St. Jude Hospital, Scholar Strategies, University of Missouri St. Louis, Ohio State University Health Services and Management Program Management Institute Annual Conference, and Nebraska Conference on Health Equity Key Note. The author is a Board Director, Vice Chair and Programmatic Strategies Chairperson for Missouri Foundation for Health, and the author receives no financial compensation for these roles. The author has no other competing interests to declare. The other authors declare that no competing interests exist.

### Funding

| Funder | Grant reference number | Author |
| --- | --- | --- |
| National Cancer Institute | P50CA244431 | Kia L Davis<br>Lisa M Klesges<br>Sarah Humble<br>Bettina Drake |
| National Institute on Minority Health and Health Disparities | T37 MD014218 | Callie Walsh-Bailey |

The funders had no role in study design, data collection and interpretation, or the decision to submit the work for publication.

### Author contributions

Kia L Davis, Conceptualization, Formal analysis, Supervision, Investigation, Methodology, Writing – original draft, Project administration, Writing – review and editing; Nicole Ackermann, Conceptualization, Data curation, Formal analysis, Investigation, Project administration, Writing – review and editing; Lisa M Klesges, Conceptualization, Writing – review and editing; Nora Leahy, Conceptualization, Investigation, Project administration, Writing – review and editing; Callie Walsh-Bailey, Writing – review and editing; Sarah Humble, Bettina Drake, Conceptualization, Methodology, Writing – review and editing; Vetta L Sanders Thompson, Conceptualization, Formal analysis, Funding acquisition, Investigation, Methodology, Writing – original draft, Writing – review and editing

## Author ORCIDs

Kia L Davis  http://orcid.org/0000-0002-1338-3018
Nicole Ackermann  http://orcid.org/0000-0001-7411-3233
Sarah Humble  http://orcid.org/0000-0003-0694-091X

## Ethics

The Washington University in St. Louis, MO Institutional Review Board approved and exempted this study (ID#202006089). Informed consent was obtained before the survey was administered. All participants received an agreed-upon incentive from Qualtrics.

## Decision letter and Author response

Decision letter https://doi.org/10.7554/eLife.85024.sa1
Author response https://doi.org/10.7554/eLife.85024.sa2

---

## Additional files

### Supplementary files

• Supplementary file 1. Survey item information.
• Supplementary file 2. Characteristics of residents across Missouri and Southern Illinois by race (July-August 2020).
• MDAR checklist
• Source code 1. Care disruption analysis SAS code file.

### Data availability

The data that support the findings of this study are openly available on Open Science Framework at https://osf.io/p5x3s/. Please cite as data from this publication if used.

The following dataset was generated:

| Author(s) | Year | Dataset title | Dataset URL | Database and Identifier |
|---|---|---|---|---|
| Kia L D, Nicole A, Lisa K, Nora L, Sarah H, Bettina D, Vetta LST | 2023 | Siteman Cancer Center COVID and cancer data | https://osf.io/p5x3s/ | Open Science Framework, p5x3s |

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
