## [Editor Report]

The study presents patterns of cancer care disruption in southern Illinois and eastern Missouri in the summer of 2020. Survey results show factors that impact cancer care during the COVID-19 pandemic, including group differences by race. The important findings provide solid evidence about variation in cancer care disruptions and opportunities to improve return to care.

---

## [Decision Letter]

**Decision letter after peer review:**

Thank you for submitting your article "Understanding disruptions in cancer care to reduce increased cancer burden: A cross-sectional study" for consideration by *eLife*. Your article has been reviewed by 2 peer reviewers, and I have overseen the evaluation in my dual role of Reviewing Editor and Senior Editor.

Essential revisions:

As is customary in *eLife*, the reviewers have discussed their critiques with one another. What follows below is an edited compilation of the essential and ancillary points provided by reviewers in their critiques and in their interaction post-review. Please submit a revised version that addresses these concerns directly. Although we expect that you will address these comments in your response letter, we also need to see the corresponding revision clearly marked in the text of the manuscript. Some of the reviewers' comments may seem to be simple queries or challenges that do not prompt revisions to the text. Please keep in mind, however, that readers may have the same perspective as the reviewers. Therefore, it is essential that you attempt to amend or expand the text to clarify the narrative accordingly.

*Reviewer #1 (Recommendations for the authors):*

The study lacks focus and significance. The methods used are not adequate for a rigorous study. There were disruptions in cancer care and other treatments during the first year of Covid (June-August 2020). While it is important to investigate whether there were disparities or inequities in access to care, the data used – a cross-sectional analysis of primary data of 650 people – is not the right data to be used to assess the disruption in cancer care.

To investigate disruption in care, I suggest using the difference-in-differences or a similar method. I would also suggest the authors use national survey data or other large datasets to conduct a trend analysis (before and after). This can provide useful information on the type of illnesses (or type of cancer treatments) or patient subpopulations that were impacted more than others by Covid – interruption in care. To conduct any type of disparity or inequity analysis, investigators need to ensure that they match subpopulations that they like to compare to one another based on their health needs. Otherwise, there are confounding variables that most certainly would bias the results.

What conceptual model did the authors use to select the explanatory variables? It seems to me that authors just throw a bunch of variables in a logistic regression model. If this study examined disruption to care, the most important explanatory variables to include in the model were the basic health status or severity and type of cancer.

The sample size was small, and with the cross-sectional nature of the study, no conclusion can be achieved. The characteristics of people with missing data were not presented and discussed. This can easily create a biased sample to begin with. The variables used and the rationale to include them have not been discussed.

*Reviewer #2 (Recommendations for the authors):*

I appreciate the opportunity to review your manuscript submitted to *eLife*. The manuscript addresses an important topic, and it will be a valuable contribution to the literature on the impact of COVID-19 on cancer care.

1. Outcome measures were adapted from an unpublished questionnaire by Penedo et al. Please provide more information about how these measures were developed and/or whether they were validated.

2. Figure 1 shows that 53 participants had appointments for cancer-related care other than screening. Because this finding is presented in Figure 1, it should be discussed in the results. A footnote in Supplemental Table 1 incorrectly states that the variable was not included in the manuscript.

3. The analysis should address the assumptions for conducting logistic regression. Specifically, authors should test for (1) linearity between continuous independent variables and the log-odds of care disruption, and (2) absence of multicollinearity. In addition, the authors treated education and income as continuous variables -- please explain this decision.

4. Cancer care disruptions were stratified by race. Did the "Patient needs for rescheduling" likewise vary based on race or other factors?

5. Please consider adding a supplemental table to show the univariate characteristics of Non-Hispanic Black or African American and Non-Hispanic White subgroups, which are stratified in the multivariate analysis in Table 2. The analysis was stratified by race because of the differential socioeconomic factors and COVID-19 impacts among communities of color -- it would be helpful to see the extent to which the combination of factors is reflected in the sample.

---

## [Author Response]

Essential revisions:As is customary in eLife, the reviewers have discussed their critiques with one another. What follows below is an edited compilation of the essential and ancillary points provided by reviewers in their critiques and in their interaction post-review. Please submit a revised version that addresses these concerns directly. Although we expect that you will address these comments in your response letter, we also need to see the corresponding revision clearly marked in the text of the manuscript. Some of the reviewers' comments may seem to be simple queries or challenges that do not prompt revisions to the text. Please keep in mind, however, that readers may have the same perspective as the reviewers. Therefore, it is essential that you attempt to amend or expand the text to clarify the narrative accordingly.Reviewer #1 (Recommendations for the authors):The study lacks focus and significance. The methods used are not adequate for a rigorous study. There were disruptions in cancer care and other treatments during the first year of Covid (June-August 2020). While it is important to investigate whether there were disparities or inequities in access to care, the data used – a cross-sectional analysis of primary data of 650 people – is not the right data to be used to assess the disruption in cancer care.To investigate disruption in care, I suggest using the difference-in-differences or a similar method. I would also suggest the authors use national survey data or other large datasets to conduct a trend analysis (before and after). This can provide useful information on the type of illnesses (or type of cancer treatments) or patient subpopulations that were impacted more than others by Covid – interruption in care. To conduct any type of disparity or inequity analysis, investigators need to ensure that they match subpopulations that they like to compare to one another based on their health needs. Otherwise, there are confounding variables that most certainly would bias the results.

Thank you for this comment. We agree that it is important to understand national trends in COVID-related care disruptions and using causal methods. However, this manuscript was not intended to be generalizable to the nation. Instead, it aimed to examine the local impact of COVID care disruptions. We focused on the Siteman Cancer Center’s (SCC) catchment area because the co-author team includes the SCC’s Associate Director of Community Outreach and Engagement (COE) program, the SCC Associate Director for Diversity, Equity, and Inclusion, multiple members of the SCC COE leadership team. Thus, we are uniquely positioned to mobilize and identify outreach opportunities and/or programs that address any gaps we discover. Moreover, this focus on our catchment area aligns with the National Cancer Institute’s priorities to characterize cancer-relevant knowledge across cancer center catchment areas. Finally, this is our first cross-sectional study of our catchment area, so we could not implement a difference-in-difference analysis. We will regularly survey residents in our catchment in the future. We will be able to track trends over time and consider additional methods like difference-in-difference once we have multiple data points. We have added text in the manuscript to emphasize the purpose of this study as a local, area-level assessment and to indicate that confounding bias is possible in cross-sectional studies, including this one.

What conceptual model did the authors use to select the explanatory variables? It seems to me that authors just throw a bunch of variables in a logistic regression model.

Our analysis was guided by the theory that social identities related to race, ethnicity, class, and gender shape access to healthcare and disease processes and are the fundamental drivers of health. We edited the sentence below in the methods section to help clarify this.

“This analysis is guided by the theory that social identities like race, ethnicity, social class, and gender shape many contextual factors related to cancer, COVID, and other health outcomes and are ultimately, the fundamental drivers of disease”.

If this study examined disruption to care, the most important explanatory variables to include in the model were the basic health status or severity and type of cancer.

We did not ask severity of cancer, but we conducted a posthoc analysis to understand care disruption among those with multiple comorbidities to understand basic health status and determine who may have additional health needs. We found that among Black residents, those that had 3+ comorbidities were more likely to have care disruptions. We have included this in the results and discussed implications in the discussion. We note that the sample size is small.

The sample size was small, and with the cross-sectional nature of the study, no conclusion can be achieved.

This was meant to be an area-level, local assessment to help Siteman Cancer Center understand how to deploy resources to mitigate the potential for health disparities. There were no causal interpretations.

The characteristics of people with missing data were not presented and discussed. This can easily create a biased sample to begin with. The variables used and the rationale to include them have not been discussed.

We checked for differential exclusion for those who were categorized as does not apply for care disruption. This sentence appears in the analytic procedures section: “We dropped those who reported that canceling an appointment did not apply to them (n=84) with more males, uninsured people, and those without telehealth appointments reflected in this exclusion.”

We also added the text below to describe the 17 people missing from the regression model. Given that this is less than 2% of our total sample, it is unlikely that this has introduced bias.

“Additionally, there were 17 residents who had missing data and were not included in the logistic regression models. This missing data was due to responding to sex assigned at birth or sexual orientation questions with “prefer not to answer” or having a missing value on another question included in the model (see footnote on Table 1 for details).”

Reviewer #2 (Recommendations for the authors):I appreciate the opportunity to review your manuscript submitted to eLife. The manuscript addresses an important topic, and it will be a valuable contribution to the literature on the impact of COVID-19 on cancer care.1. Outcome measures were adapted from an unpublished questionnaire by Penedo et al. Please provide more information about how these measures were developed and/or whether they were validated.

Since our article was submitted, the questionnaire has been published. The questions were drawn from validated measures assessing the impact of pandemics such as H1N1, and major life disruptions such as natural disasters. This language was updated in the manuscript as were the references.

Relevant Reference: Saez-Clarke, E., Otto, A. K., Prinsloo, S., Natori, A., Wagner, R. W., Gomez, T. I., … and Penedo, F. J. (2023). Development and initial psychometric evaluation of a COVID-related psychosocial experiences questionnaire for cancer survivors. *Quality of Life Research*, 1-20.

2. Figure 1 shows that 53 participants had appointments for cancer-related care other than screening. Because this finding is presented in Figure 1, it should be discussed in the results. A footnote in Supplemental Table 1 incorrectly states that the variable was not included in the manuscript.

We renamed this Supplementary file 1 as directed by *eLife* editors. We also added the language below to the Results section.

Approximately 53 people in the sample reported having cancer. Among those, 26% reported having to cancel/postpone their cancer-related care. We also corrected the table to add that this is included in the paper, but not included in the outcome measure.

3. The analysis should address the assumptions for conducting logistic regression. Specifically, authors should test for (1) linearity between continuous independent variables and the log-odds of care disruption, and (2) absence of multicollinearity.

We tested for multicollinearity by calculating variance inflation factors. All values were less than 2, indicating no evidence of multicollinearity. Additionally, a correlation matrix of all intendent variables in the model showed no values over 0.8 (indicative of a high correlation and potential for multicollinearity); all correlations were under 0.5.

In addition, the authors treated education and income as continuous variables -- please explain this decision.

We have 5 education categories and 8 income categories. We decided to model them as continuous given the multiple categories and to increase power.

4. Cancer care disruptions were stratified by race. Did the "Patient needs for rescheduling" likewise vary based on race or other factors?

In a supplemental analysis, we stratified patient care needs by race. We have added the test below to our results. Results were mostly similar.

Results stayed similar when looking at the highest need for rescheduling by race, except that the third highest need for Black residents was time to schedule the appointment (14.2%), and for White residents, it was dealing with other things and not ready to reschedule yet (14.5%). The top two needs to reschedule appointments were the same across race though the proportion of those who selected each need varied somewhat by race. The top two needs were knowing if their doctor’s office or clinic is taking the appropriate COVID-related safety precautions (Non-Hispanic Black: 34.2%; Non-Hispanic White: 33.8%; All Other: 32.7%) and not needing anything for all three racial groups (Non-Hispanic Black: 19.2%; Non-Hispanic White: 18.8%; All Other: 12.2%).

5. Please consider adding a supplemental table to show the univariate characteristics of Non-Hispanic Black or African American and Non-Hispanic White subgroups, which are stratified in the multivariate analysis in Table 2. The analysis was stratified by race because of the differential socioeconomic factors and COVID-19 impacts among communities of color -- it would be helpful to see the extent to which the combination of factors is reflected in the sample.

Thank you for this suggestion. In the newly added Supplementary file 2, we show the variables of interest by race strata.